# *Padina boergesenii*-Mediated Copper Oxide Nanoparticles Synthesis, with Their Antibacterial and Anticancer Potential

**DOI:** 10.3390/biomedicines11082285

**Published:** 2023-08-17

**Authors:** Thirupathi Balaji, Chethakkad Manikkan Manushankar, Khalid A. Al-Ghanim, Chinnaperumal Kamaraj, Durairaj Thirumurugan, Sundaram Thanigaivel, Marcello Nicoletti, Nadezhda Sachivkina, Marimuthu Govindarajan

**Affiliations:** 1Department of Biotechnology, Faculty of Science and Humanities, SRM Institute of Science and Technology, Chengalpattu Dt., Kattankulathur 603203, Tamil Nadu, India; balajieinbiotech@gmail.com (T.B.); cmmanusankar@gmail.com (C.M.M.); thanigaivel092@gmail.com (S.T.); 2Department of Zoology, College of Science, King Saud University, Riyadh 11451, Saudi Arabia; kghanim@ksu.edu.sa; 3Directorate of Research and Virtual Education, Interdisciplinary Institute of Indian System of Medicine (IIISM), SRM Institute of Science and Technology (SRMIST), Chengalpattu Dt., Kattankulathur 603203, Tamil Nadu, India; kamarajc@srmist.edu.in; 4Department of Environmental Biology, Foundation in Unam Sapientiam, Sapienza University of Rome, 00185 Rome, Italy; marcello.nicoletti@uniroma1.it; 5Department of Microbiology V.S. Kiktenko, Institute of Medicine, Peoples Friendship University of Russia Named after Patrice Lumumba (RUDN University), Moscow 117198, Russia; 6Unit of Vector Control, Phytochemistry and Nanotechnology, Department of Zoology, Annamalai University, Annamalainagar 608 002, Tamil Nadu, India; drgovind1979@gmail.com; 7Unit of Natural Products and Nanotechnology, Department of Zoology, Government College for Women (Autonomous), Kumbakonam 612 001, Tamil Nadu, India

**Keywords:** nanotechnology, green synthesis, CuO nanoparticle, microbial biofilms, biomedical applications, *Padina boergesenii*, A375 cell line

## Abstract

The utilization of nanoparticles derived from algae has generated increasing attention owing to their environmentally sustainable characteristics and their capacity to interact harmoniously with biologically active metabolites. The present study utilized *P. boergesenii* for the purpose of synthesizing copper oxide nanoparticles (CuONPs), which were subsequently subjected to in vitro assessment against various bacterial pathogens and cancer cells A375. The biosynthesized CuONPs were subjected to various analytical techniques including FTIR, XRD, HRSEM, TEM, and Zeta sizer analyses in order to characterize their stability and assess their size distribution. The utilization of Fourier Transform Infrared (FTIR) analysis has provided confirmation that the algal metabolites serve to stabilize the CuONPs and function as capping agents. The X-ray diffraction (XRD) analysis revealed a distinct peak associated with the (103) plane, characterized by its sharpness and high intensity, indicating its crystalline properties. The size of the CuONPs in the tetragonal crystalline structure was measured to be 76 nm, and they exhibited a negative zeta potential. The biological assay demonstrated that the CuONPs exhibited significant antibacterial activity when tested against both *Bacillus subtilis* and *Escherichia coli*. The cytotoxic effects of CuONPs and cisplatin, when tested at a concentration of 100 µg/mL on the A375 malignant melanoma cell line, were approximately 70% and 95%, respectively. The CuONPs that were synthesized demonstrated significant potential in terms of their antibacterial properties and their ability to inhibit the growth of malignant melanoma cells.

## 1. Introduction

The incidence of skin cancer has witnessed a notable rise in recent years [1], thereby necessitating the development of innovative treatment approaches. Contemporary therapeutic modalities are currently constrained to chemotherapy, surgical resection, and radiotherapy. Unfortunately, the efficacy of these treatments is only moderate, and they are associated with adverse side effects. Hence, there is a pressing need for the development of novel therapeutic agents that exhibit reduced or negligible side effects. Natural products have been extensively employed in pharmacological research and have demonstrated significant biological activities in the treatment of cancer, often exhibiting reduced toxicity compared to other therapeutic approaches [2]. The distinctive physical, optical, chemical, and mechanical properties of nano-biomaterials have contributed to their rapid rise in popularity over the last ten years. In addition, these platforms provide a method for encapsulating pharmaceuticals that have poor solubility, which makes it easier for the medications to be delivered to the body while also extending the drug’s half-life [3]. The formation of harmful byproducts may be kept to a minimum via the use of biological processes that are both cost-effective and safe [4,5]. This process can be used to create biocompatible nanoparticles. Copper oxide nanoparticles, often known as CuONPs, are one kind of nanoparticle that has a wide variety of uses in the area of pharmacology.

Electrochemical processes, solvent flame-based synthesis employing microwave-assisted technology, co-precipitation, mechanical mixing, the process of hydrolysis, and the sol-gel technique are some of the many ways that may be utilized for producing CuNPs [6,7]. On the other hand, a number of studies have shown that NPs manufactured using procedures that are not biological cannot be used in clinical settings. The synthesis of CuNPs using biomolecules as reducing and stabilizing agents has repeatedly been shown to be a fruitful technique [8]. In addition, CuONPs have attracted much interest because of the increased surface area and greater stability that they possess [8]. One of the most significant challenges that the healthcare sector is presently facing is the proliferation of illnesses that are unresponsive to the conventional therapies that have been developed [9]. CuONPs have the potential to have a physiological impact because of their anticancer, antibacterial, anti-biofilm, anti-inflammatory, and free radical fighting abilities [10,11,12].

In particular, the CuONPs play an essential position in signaling pathways and cause DNA damage as a consequence of their exposure [13]. Additionally, it has been found that this mechanism also induces apoptosis inside live cells by producing reactive oxygen species [14]. This occurs as a result of the creation of reactive oxygen species. Recent research has shown that the metabolic products of algae have the potential to aid in the creation of metal NPs [12,13]. These findings were published in two separate investigations. Earlier research [15,16,17] employed metabolites originating from seaweed as a means of biologically manufacturing nanoparticles of copper, iron, zinc, gold, silver, and selenium. These nanoparticles were produced in seaweed’s metabolites. CuNPs are now being used in a variety of industries, including agriculture, the textile industry, and the dye industry [11,17,18]. This is a result of the fact that CuNPs are not dangerous, their production costs are low, and they have a high level of stability.

Brown seaweed *Padina boergesenii* (Dictyotaceae family, Phaeophyta) is regularly occurring along India’s southeast coast in the Gulf of Mannar. High levels of phenolics and antioxidant activity were reported [19]. The synthesis, characterization, and inhibitory effects of *P. boergesenii* aqueous extract on glucosidase in in vitro animal models were described by Senthilkumar et al. [20,21]. The potential of this extract in treating hyperglycemia was also investigated [22]. Moreover, previous studies have provided evidence of hepatoprotective action and chemopreventive effects [23,24]. According to epidemiological studies, the consumption of a diet rich in seaweed has been associated with a potential decrease in the incidence of cancer and other notable diseases [25]. The genus *Padina*, comprising brown algae, possesses a substantial reservoir of biologically active metabolic end products [26]. These end products exhibit the ability to function as both a reductant and a capping agent for the synthesis of nanoparticles. The bioactivity of the algal sample has been extensively studied, and some nanoparticles’ syntheses have been described in *Padina* sp. Further research is needed to investigate the combined effects of metal precursors and the extract [5,20]. The current study has been designed to synthesize CuONPs along with the aqueous extract of *P. boergesenii* and test the efficacy of their antibacterial, anti-proliferative, and apoptosis-inducing capacity on the A375 skin carcinoma cell line.

## 2. Materials and Methods

### 2.1. Seaweed Processing and Preparation

The brown alga *Padina boergesenii* Allerder & Kraft, which had recently been collected, was acquired from the coastal area of Mandapam in the Gulf of Mannar, located at a latitude of 9°17′ N and a longitude of 79°11′ E, in the state of Tamilnadu, India. To mitigate the risk of contamination, the specimens underwent a purification process using distilled water, followed by their placement in polythene bags for the purpose of the study. The alga underwent a process of desiccation and subsequent pulverization using an electric blender. The powdered sample was stored in a light-restricted environment and subsequently utilized for subsequent investigations. In order to acquire the seaweed extract, 10 g of powdered seaweed was submerged in 100 milliliters of sterile deionized water. The mixture was vigorously stirred and subsequently filtered through a No. 1 paper filter. The filtrate was subjected to refrigeration and preservation for subsequent research [27].

### 2.2. Biological Synthesis of CuONPs

The synthesis of CuONPs was conducted by combining 10 mL of the aqueous filtrate of *P. boergesenii* with 90 mL of a copper sulphate solution in various ratios (3:7, 2:8, and 1:9). The mixture was then stirred at room temperature for a duration of 30 min. The pH value of 7.0 was modified by the introduction of a 1N NaOH solution. Additionally, a distinct green precipitate was observed in the solution resulting from the mixture of substances in a 1:9 ratio. The mixture was subsequently subjected to centrifugation, and the resulting pellets were subjected to overnight drying at a temperature of 50 °C. To facilitate additional examinations, the pellets were stored at room temperature.

### 2.3. Characterization of CuONPs

Different analytical techniques were utilized to characterize the synthesized NPs in order to assess their functionality. The functional groups of producing CuONPs were investigated using an FT-IR spectrometer (Bruker Scientific, Billerica, MA, USA). Using an X-ray diffractometer for powder (Bruker Scientific, Billerica, MA, USA), the crystalline structure of the produced CuONP was examined. The particle size of the nanoparticle was determined by zeta potential and a particle analyzer. High-resolution scanning electron microscopy (HRSEM) and an Energy Dispersive X-ray Spectrometer (EDS) were used to evaluate the elemental composition of CuONPs (Thermo Fisher Scientific, Waltham, MA, USA). High-resolution transmission electron microscopy JEM-2100 (HRTEM) (JEOL, Tokyo, Japan) was used to examine the structural characteristics of NPs.

### 2.4. Antibacterial Study

The antibacterial effectiveness of synthesized CuO nanoparticles was evaluated using the agar well diffusion method. The antibacterial assay employed pathogenic bacteria, such as *Escherichia coli* (MTCC 443), *Pseudomonas aeruginosa* (MTCC 424), and *Bacillus subtilis* (MTCC 5981). CuONPs were produced at two-fold concentrations (50, 100, and 200 μg/mL). Measurements of inhibitory zones were made after 24 h of incubation by measuring the width of inhibitory zones around the wells.

### 2.5. Anticancer Study

The vitality and cytotoxicity of A375 cells exposed to various doses of bio-synthesized CuONPs and cisplatin were assessed using the MTT test. Trypsinization of the monolayer cell cultures was performed in a medium containing 10% fetal bovine serum (FBS) and densities were increased to 1 × 10^5^ cells/mL. An aliquot of the diluted cell suspension was put in each well and incubated for 24 h. Once a partial monolayer had formed, the supernatant was decanted, and the cells were given one medium wash. Following that, 100 μL of varied concentrations were applied to the cells (6.25–100 µg/mL) of biosynthesized CuONPs and cisplatin and kept at 37 °C in 5% CO_2_ for 24 h. After discarding the test solutions in the wells and incubating with 5% CO_2_, 20 µL of MTT (2 mg/mL in the PBS solution) was added to each well. Following precipitate removal, 100 mL of DMSO was used to solubilize the formazan. After each treatment, the optical density at 570 nm was determined in triplicate using a spectrophotometric technique [28]. The result percentages were calculated using control untreated cells as a standard. In order to calculate the percentage of viability (%), the following formula was used: % of viability = Sample abs/Control abs × 100. The IC_50_ values of CuONPs and cisplatin on A375 cell lines were calculated.

### 2.6. Statistical Methods

All data were displayed with their respective means and standard deviations. Analysis of variance and the Bonferroni test for multiple comparisons, both with a significance threshold of *p* < 0.05, were used in the statistical analyses. 

## 3. Results and Discussion

The present advancements in nanotechnology hold promise for enhancing the antibacterial and anticancer efficacy of phytochemicals. The present analysis places specific emphasis on the cytotoxicity and antibacterial characteristics of CuONPs. Metal nanoparticles with desired characteristics could be synthesized by exploiting the inherent resistance of brown algae to heavy metals such as copper and lead [29]. Thus, the *P. boergesenii* was screened for the biosynthesis of CuONPs. One of the most recent biological techniques is the “green synthesis method”, which uses biochemicals, i.e., enzymes, vitamins, and polysaccharides in plants, bacteria, fungi, algae, and plants [30,31,32,33,34,35], as well as microorganisms [36,37]. The biological approach that makes use of plant extracts is a rapid and efficient strategy for producing different metal nanoparticles without the need for risky, expensive, and harmful components [5,38,39]. Flavonoids, alkaloids, terpenoids, phenolic acid, and tannins are some of these secondary metabolites, which are essential for the bioremediation of metal ions and advantageous in the production of various nanoparticles [40,41].

CuONPs were successfully synthesized by employing an aqueous extract obtained from *P. boergesenii*, in combination with copper sulphate. The formation of CuONPs was confirmed through ultraviolet spectral analysis, which revealed a noticeable change in color from blue to green. This color shift can be attributed to the reaction that occurred between the aqueous extract and copper solution. A similar change in hue resembling the color green was observed during the production of CuONPs using a plant extract [25] and macroalgae [28,29]. The studies mentioned above primarily focused on investigating the use of plant and algal extracts as reducing agents, without conducting a detailed analysis of specific metabolites in either study. The production of CuONPs was identified through an investigation of the color change response using a UV-visible spectrophotometer. The surface plasmon resonance at 274 nm was observed, as depicted in Figure 1.

Figure 2 shows the FT-IR spectra of CuONPs and extracts of *P. boergesenii*. The FT-IR band at 3065 cm^−1^ is attributable to the asymmetric stretching vibration of C-H (alkanes). The band represents the amide C=C, C=O stretching in the proteins at 1791 cm^−1^. The weak FT-IR band at 1376 cm^−1^ was connected to the C-H bending and C-N stretching. The stretching of C-O (alcohol and ether groups) is due to the strong and distinct FT-IR signal at 1087 cm^−1^. An additional FT-IR rise at 623 cm^−1^ can be attributed to metal-C=O stretching (metal ions interacting with the carboxylic group). Due to the aqueous extracts of *P. boergesenii*, a second FT-IR signal at 3464 cm^−1^ can be attributed to the OH-stretching vibration (Figure 2). Comparable FT-IR bands were observed in the synthesized CuONPs. The presence of alkanes, carboxylic, alcohol, ether, and hydroxyl functional groups was reduced and shifted in the CuONPs due to the interaction with NPs. The aqueous extracts of *P. boergesenii* contain bio-organic components that exhibit a strong affinity for Cu^2+^ ions, primarily through electrostatic interactions. The reduction of Cu^2+^ to CuONPs within the biological matrix can be facilitated by bio-organic constituents, which also serve as stabilizing agents for the synthesized nanoparticles. The vibrational modes associated with the C=O stretching of the peptide bond and the C-N stretching of amide II are indicated by the absorption bands observed at wavenumbers of 1638 and 1540 cm^−1^, respectively. The presence of the signal at a wavenumber of 620 cm^−1^ can be attributed to the bending of the CuO bond [17].

The XRD pattern of bio-synthesized CuONPs shows the existence of crystalline structure. The Bragg’s XRD peaks at 2 theta angles found six diffraction signals at 17.67, 28,19, 31.15, 35.78, 36.2 and 53.09, corresponding to lattice planes of (101), (112), (103), (202), (004), and (312) (Figure 3). This is in accordance with the tetragonal crystalline nature of CuONPs (JCPDS card file no. 01-071-0251). The further unsorted peaks correspond to the amorphous nature of CuONPs. A prominent peak corresponding to the (103) plane in XRD analysis is sharp, and the high intensity represents the crystalline nature of CuONPs. A Debye–Scherrer calculation (Scherrer’s equation) using the formula D = Kλ/βcosθ was used to determine the size of the CuONPs after they have been synthesized. The Bragg angle and the line broadening at half the maximum intensity (FWHM) are included in the formula where D is the diameter of NPs and λ is the wavelength for the X-ray. Thus, the average crystalline size of CuONPs was measured here to be 9.76 nm. In a similar study, CuONPs synthesized from the *Bougainvillea* flower were reported to have an average particle size of 12 nm [42].

The confirmation of the size, structure, and appearance of CuONPs was achieved through the utilization of FE-SEM and TEM studies. Figure 4 presents a scanning electron microscopy (SEM) image accompanied by energy-dispersive X-ray spectroscopy (EDX) analysis of the synthesized copper oxide nanoparticles (CuONPs). The analysis using scanning electron microscopy (SEM) reveals the existence of a tetragonal crystalline morphology, with certain particles exhibiting agglomeration. This agglomeration can be attributed to the adhesive properties of the *P. boergesenii* extract, as mentioned in reference [43]. The analysis conducted using EDX demonstrated the identification of copper and oxide elements within the nanoparticle, thereby confirming the formation of CuONPs. The CuONPs were synthesized through the process of electrostatic interactions and the coupling of bioactive capping molecules [44].

The electron micrograph in Figure 5 presents an illustration of CuONPs along with energy-dispersive X-ray spectroscopy (EDS) analysis, which provides confirmation of the presence of copper and oxide elements. Additionally, the micrograph allows for the examination of the structure, size, shape, and purity of the CuONPs. The transmission electron microscopy (TEM) images depict the tetragonal crystalline structure, exhibiting a size range of 22 to 159 nm (Figure 5A). The research findings indicated a broad distribution of particles, with an average size of 83 nm (Figure 5D). The phenomenon of nanoparticle agglomeration in certain regions has been noted, potentially attributed to the interactions between surfactant chains within SDS molecules. The researchers observed that the size of CuONPs derived from seaweed *S. longifolium* ranged from 40 to 60 nm [45]. The confirmation of the chemical composition and stability of CuO nanoparticles was achieved through the utilization of EDS measurements. The analysis of the particle composition demonstrated that CuONPs consisted of 75% copper (Cu) and 25% oxygen (O) elements, as depicted in Figure 5C. The robust signals observed indicate the existence of elemental copper (Cu) and oxygen (O), which combine to form pure copper oxide nanoparticles (CuONPs). The identification of metabolites in algal extracts is determined by the presence of copper (Cu) and oxygen (O), along with minor traces of oxygen and carbon contamination observed in proximity to the peaks [45]. Researchers have revealed that depending on the reducing and capping agents utilized, biologically produced CuNPs may have asymmetric forms or contain various nanostructures, including hexagonal, cylindrical, triangular, and prismatic morphologies [46,47,48].

The present study aimed to investigate the antibacterial properties of CuONPs at different concentrations (50, 100, and 200 µg/mL) and ampicillin (25 µg/mL) against a panel of bacterial pathogens. The synthesized CuONPs derived from *P. boergesenii* demonstrated notable antibacterial efficacy against *B. subtilis* (10.33 ± 0.57 mm), *E. coli* (12.33 ± 0.57 mm), and *P. aeruginosa* (12.33 ± 0.57 mm) at a concentration of 200 μg/mL (Figure 6 and Table 1). The antimicrobial efficacy of CuONP is significant in combating bacterial strains belonging to both the Gram-negative and Gram-positive categories [49]. The CuONP exhibits significant inhibitory activity against *E. coli* (15 mm), *B. subtilis* (11 mm), and *P. aeruginosa* (21 mm) at a concentration of 200 μg/mL [49]. The observed inhibition could potentially be attributed to a reduction in the integrity of the bacterial membrane. Copper nanoparticles have the potential to decrease the electrochemical potential of the bacterial transmembrane, thereby posing a risk to the membrane’s integrity [50].

The copper nanoparticles (CuNPs) synthesized using the brown alga *Sargassum vulgare* exhibited potential antibacterial activity and demonstrated efficacy in inhibiting biofilm formation caused by *Staphylococcus aureus*. The minimum inhibitory concentration (MIC) values for CuNPs were determined to be 250 and 150 μg/mL for methicillin-resistant *S. aureus* (MRSA) and methicillin-sensitive *S. aureus* (MSSA), respectively. The study reported that *S. aureus* exhibited significant antibiofilm activity at concentrations of 100 and 50 μg/mL [51]. In their study, Ramaswamy et al. [52] examined the synthesis of copper oxide nanoparticles using *S. polycystum*. The findings indicated that the antimicrobial activity of these nanoparticles was the least pronounced against *Shigella dysenteriae*, with an average inhibition zone of 6 ± 0.5 mm. Conversely, the highest antimicrobial activity was observed against *Pseudomonas aeruginosa*, with an average inhibition zone of 15 ± 0.5 mm.

Previous research conducted by Bhacuni [53] discovered that CuNPs derived from *Tritium aestivum* exhibited a maximal inhibitory zone of 18 mm against *Aspergillus niger*, which aligns with the findings of the current study. A minimum inhibitory activity of 9 mm zone was also reported for *A. flavus*. The antibacterial efficacy of Mithun urine-mediated CuO NPs was evaluated against *Aeromonas hydrophila* and *A. veronii*, two bacterial pathogens that are recognized for their ability to induce severe infectious diseases in fish. The two fish parasites in question share a common designation, namely Aeromonas. At a concentration of 100 µg/mL, the MU-CuONPs exhibit heightened antibacterial efficacy against two bacterial strains commonly associated with fish [54]. Moreover, in addition to *Sesbania grandiflora,* CuONPs have been found to augment the potent antibacterial properties of CuONPs against a range of pathogens, including Gram-negative bacteria such as *E. coli* and *P. aeruginosa*, as well as Gram-positive bacteria such as *S. aureus*. The growth of *E. coli* was observed to be inhibited by Sg-CuO nanoparticles at doses of 100 and 125 µg, resulting in the formation of zones of inhibition with diameters measuring 10 ± 1.21 mm and 14 ± 1.39 mm, respectively. The results indicated a significant difference (17 0.46 mm) in favor of the treatment being compared to streptomycin [55]. The efficacy of copper nanoparticles derived from *N. cataria* as bactericidal agents. The nanoparticles (NPs) exhibited the greatest level of inhibition against *E. coli* when tested at a concentration of 100 µg mL^−1^. This was followed by *Enterococcus faecalis* and *S. aureus*. The results demonstrated that *E. coli* exhibited an inhibition zone measuring 30 mm, while *E. faecalis* displayed an inhibition zone measuring 21 mm. Additionally, *S. aureus* exhibited an inhibition zone measuring 11 mm [56].

Several investigations have shown that the particle size and capping agents of CuNPs can influence antibacterial activity [11,57]. According to Bogdanovi et al. [58], the MIC value for CuNPs, chemically synthesized using sodium borohydride, have an antibacterial activity of 32 μg/mL against *S. aureus*, MRSA, and MSSA. It is thought that the antibacterial effects of CuNPs are due to their powerful binding to amine and carboxyl functions on the cell surfaces of Gram-positive bacteria [59,60]. Copper ions will interact with DNA fragments to decrease proliferation and gene expression in the bacterial cell [61]. This research found that CuONPs were efficient against *P. aeruginosa* and other Gram-positive and Gram-negative bacteria. CuNPs are bactericidal against a range of bacteria, fungi, and parasites [58,62].

CuONPs and cisplatin’s cytotoxic effects at various concentrations (6.25, 12.5, 25, 50, and 100 µg/mL) were investigated on the A375 cancerous cell line. The cytotoxic effects of CuONPs and cisplatin in 100 µg/mL concentrations were found to be approximately 70% and 95%, respectively, on the human malignant melanoma (A375) cell line. The cytotoxicity resulted in being dose-dependent (Figure 7), and a statistically significant variation in the vitality of A375 cells exposed to varied doses of CuONPs was observed (*p* < 0.001). (Figure 8). The IC_50_ value of CuONP was 54.86 μg/mL, which caused 50% cell death. An algal-derived bimetal (ZnO-CuO) nanoparticle showed 100% cytotoxicity on the A375 cell line at a concentration of 500 µg/mL [63]. In the current study, we found that the effects of CuONPs on anticancer cell lines were significantly higher. The anticancer activity of CuONPs may be attributed to the adsorbed active molecules found in algal extracts. The toxicity of CuONPs might be mediated through a Trojan horse-like mechanism [64]. There was no change in the morphology of the cells without any treatment.

When CuONPs with the highest concentration might accumulate inside the cells, this could cause severe oxidative stress and kill cancer cells. The algal-mediated CuONPs could have more effect and better anticancer efficacy on malignant melanoma cells. However, there is no report on CuONPs from seaweed against A375 malignant melanoma cancer cell lines. Ramaswamy et al. [52] used the MTT assay to determine the IC_50_ value for copper nanoparticles, which was 61.25 μg/mL, for brown algae-mediated CuO nanoparticles against MCF-7 cell lines. More than 93% of a cell’s growth was effectively inhibited by 100 μg/mL of copper nanoparticles. Similar to this, Ramasubbu et al. [55] showed that treatment of HepG2 cells with copper nanoparticles has anticancer effects.

## 4. Conclusions

The current study suggests a successful method for synthesizing CuONPs by utilizing an aqueous extract derived from *P. boergesenii*. The selected algae extract has undergone a suitable reduction process, thereby facilitating the synthesis of nanoparticles. The findings from the FTIR analysis indicated that the presence of alcoholic or phenolic compounds in seaweed extract could potentially induce a transformation of copper sulphate into CuONPs, thereby leading to the stabilization of said nanoparticles. The study revealed that the CuONPs synthesized through biological means were enveloped by proteins, suggesting their potential as cytotoxic agents against A375 cancer cells and bacterial pathogens. This report presents novel findings on the biosynthesis of CuONPs utilizing aqueous extracts derived from *P. boergesenii*. These nanoparticles hold promise as a potential therapeutic agent for cancer treatment. However, further in vivo investigations are necessary to provide a comprehensive understanding of the actual anticancer effectiveness of these nanoparticles.

## Figures and Tables

**Figure 1 biomedicines-11-02285-f001:**
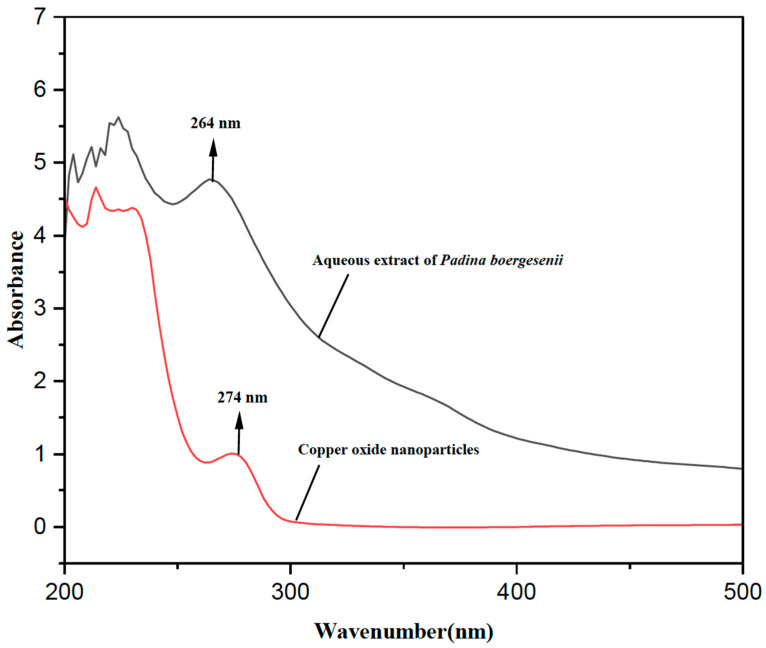
UV-visible spectrum of *P. boergesenii* extract and bio-synthesized copper oxide nanoparticle (CuONPs).

**Figure 2 biomedicines-11-02285-f002:**
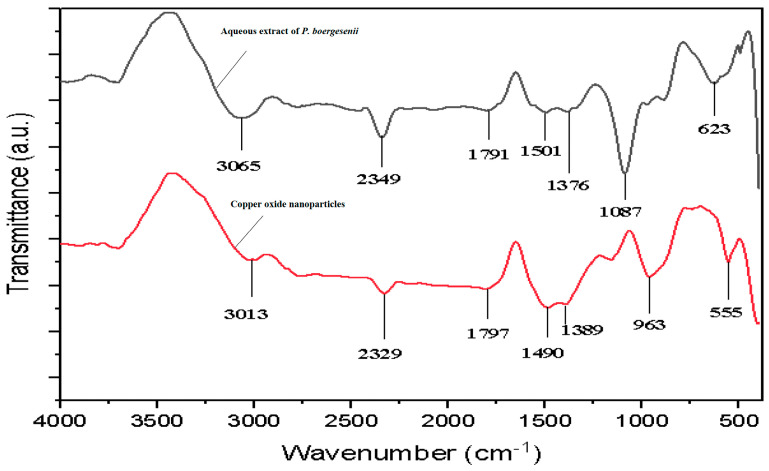
FT−IR spectrum of *P. boergesenii* extract and synthesized Copper oxide Nanoparticles.

**Figure 3 biomedicines-11-02285-f003:**
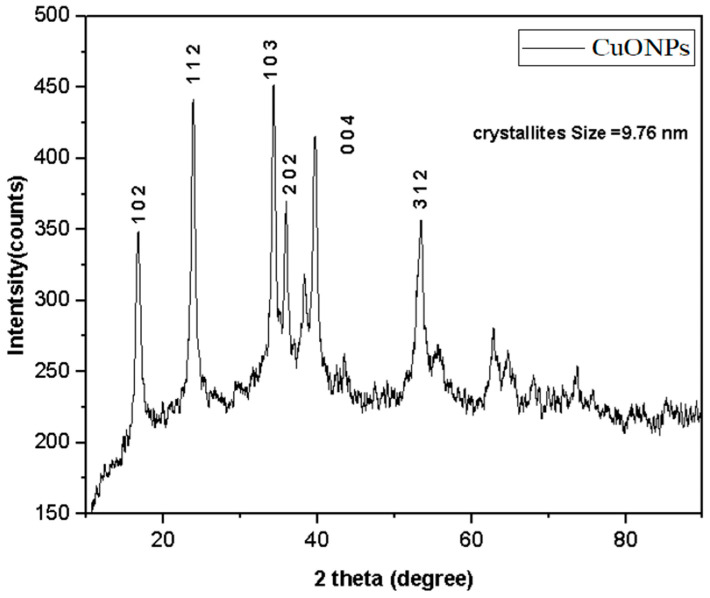
XRD pattern of CuONPs synthesized from *P. boergesenii*.

**Figure 4 biomedicines-11-02285-f004:**
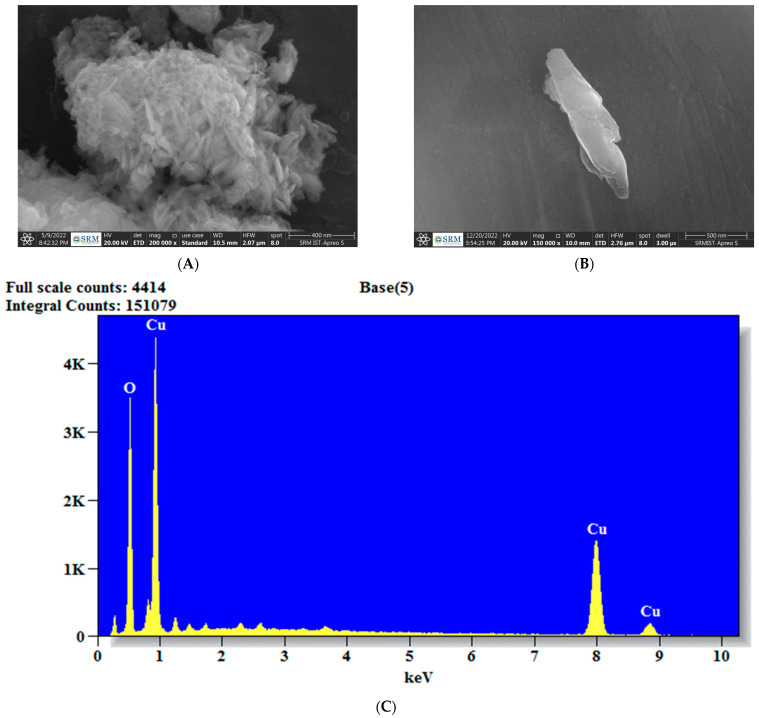
(**A**) Morphological view of nanoparticles at 200,000× magnification. (**B**) Morphological view of nanoparticles at 150,000× magnifications. (**C**) Energy-Dispersive X-ray spectra of CuONPs. K = 1000.

**Figure 5 biomedicines-11-02285-f005:**
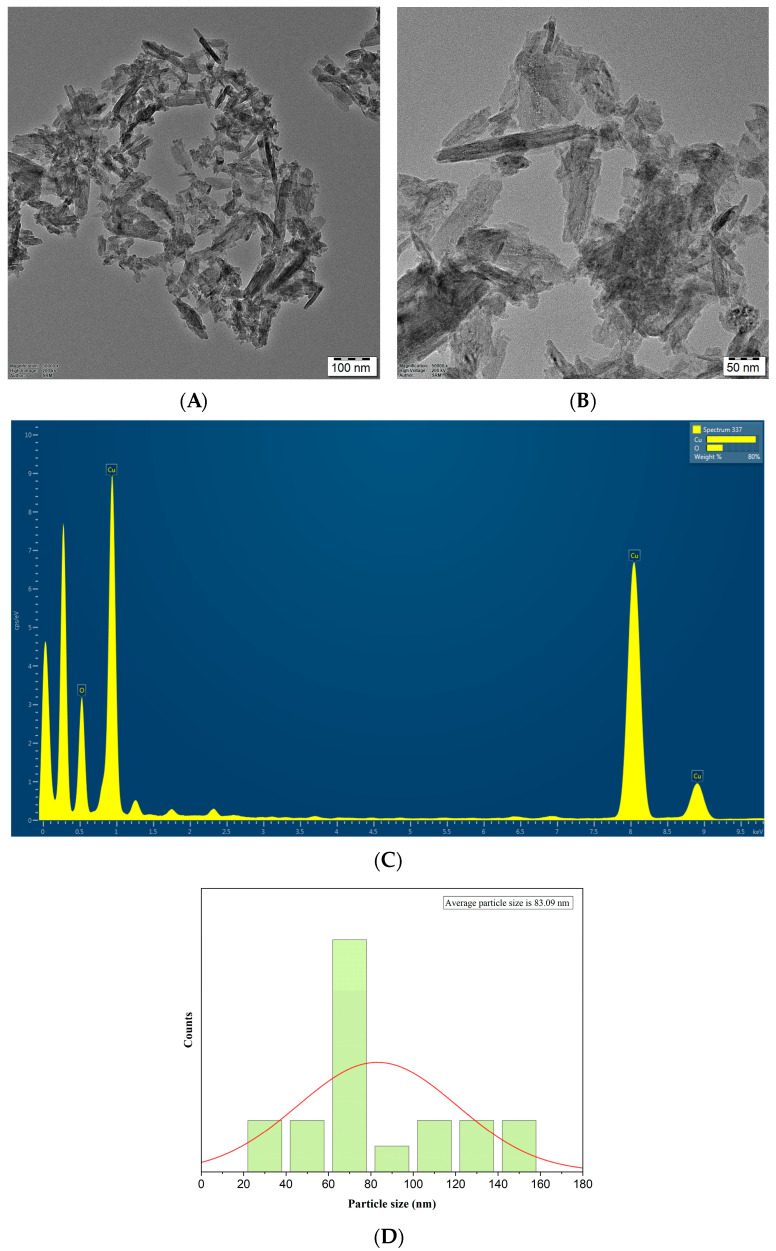
(**A**,**B**) TEM images (**C**) EDS image and (**D**) Particle size distribution histogram of CuONPs.

**Figure 6 biomedicines-11-02285-f006:**
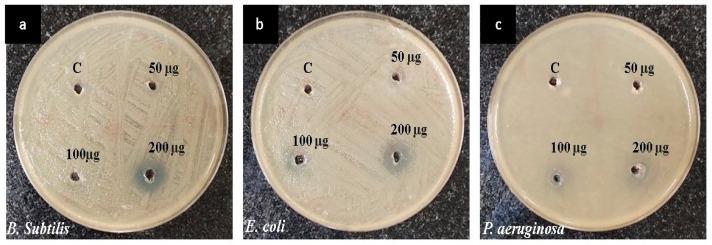
Antibacterial activity of CuONPs against (**a**) *B. subtilis*, (**b**) *E. coli,* and (**c**) *P. aeruginosa*.

**Figure 7 biomedicines-11-02285-f007:**
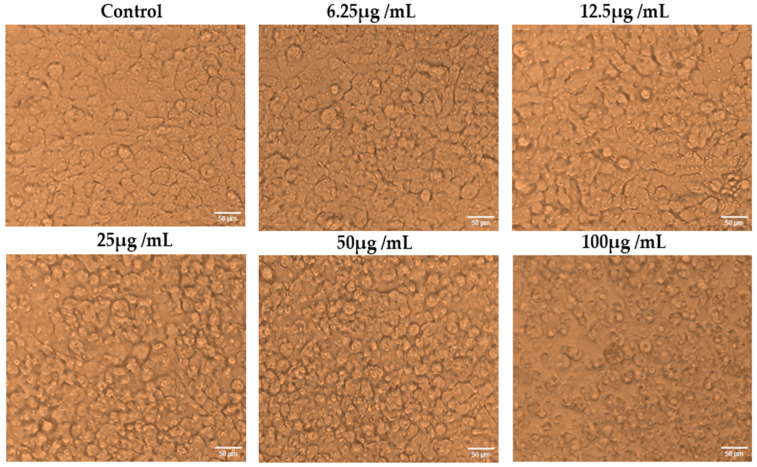
Cytotoxic activity of bio-synthesized CuONPs against A375 cell line.

**Figure 8 biomedicines-11-02285-f008:**
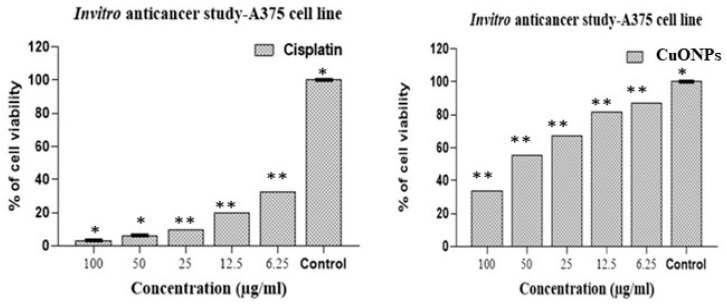
Effects on the viability of A375 cells exposed to different concentrations of bio-synthesized CuONPs and cisplatin. (*—significance level of *p* < 0.05, **—no significant).

**Table 1 biomedicines-11-02285-t001:** Antibacterial activity of different concentrations of CuONPs against *E. coli*, *B. subtilis*, and *P. aeruginosa*.

Concentrations	Zone of Inhibition (mm)		
	*E. coli*	*B. subtilis*	*P. aeruginosa*
Control	3.33 ± 0.57	4.66 ± 0.57	3.33 ± 0.57
50 µg	3.33 ± 0.57	3.33 ± 0.57	4.66 ± 0.57
100 µg	6.33 ± 0.57	7.33 ± 0.57	6.33 ± 0.57
200 µg	12.33 ± 0.57	10.33 ± 0.57	12.33 ± 0.57

## Data Availability

Data will be made available upon request.

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
