# Peer review of "Padina boergesenii-Mediated Copper Oxide Nanoparticles Synthesis, with Their Antibacterial and Anticancer Potential"

_biomedicines, 2023, doi:10.3390/biomedicines11082285_

Round 1

Reviewer 1 Report

In this manuscript, the authors prepared copper oxide nanoparticles using padina boergesenii and evaluated their possible use in fighting bacteria and cancer cells. In general, the idea presented in the manuscript is straight forward. However, the experiment designs were not appropriate. In addition, the manuscript and data were not well-prepared. For the above reasons, it could not be recommended for publication in the journal.

Comments:

1.       Lines 98-100: The sentences should be re-written.

2.       Line 113: specific amount of P. boergesenii should be added.

3.       The mechanism for the formulation of CuONPs as prepared in section 2.2 should be explained and added in the main text.

4.       Figures 1 and 3. UV-vis spectrum of Padina boergesenii extract should also be added to compare.

5.       Figures 2, 9. What is the difference between CuONPs and PBCNPs?

6.       Figure 4’s caption: What is the meaning of the morphological view of nanoparticles at 400nm, 500nm?

7.       Lines 249-251: “SEM analysis represents the presence of a tetragonal crystalline shape, and some are agglomerated, which might be due to improper sample preparation”. A SEM image with proper sample preparation should be added.

8.       Error in the sentence in line 239.

9.       Lines 127-128: “The CuONPs' size distribution and charge were investigated using dynamic light scattering (zeta sizer, Malvern/Nano ZS-90)” and lines 251-252: “The nanoparticles were measured to be between 23 and 168 nm in size”, and figure 6A: Since the shape of the NPs was not spherical, the measurement of size using zetasizer was incorrect. The size should be estimated from TEM images of many single particles. The authors should fix this size information.

10.   Figure 7: Quantitative data of antibacterial activity of the prepared NPs should be added.

11.   Figure 8: Image quality of the cells was too low. In addition, scale bars should be added.

 Extensive editing of English language is required.

Author Response

Dear Reviewer! Thank you so much for paying attention to our work and spending your time. Our team very much appreciates your edits in the article and of course we will take them into account. We are sure that working together will only make the article better. We have tried to answer all your questions

Reviewer 2 Report

The title is not well worded and needs to be corrected.
It needs to be better aligned with the purpose and content of the manuscript.
For example: Padina boergesenii mediated synthesis of copper oxide nanoparticles with promising antibacterial and anticancer activity, etc.

The concluding sentence in the abstract must refer to the entire work, not just one biological test. Also, the IC50 value does not need to be explained in the abstract.

I suggest that the previous use of these brown algae for the synthesis of nanoparticles be mentioned in the Introduction.

It is necessary to create a new Discussion section or combine the results and the discussion in one chapter.
Also, it is necessary to improve the discussion in the light of the current knowledge on the research topic.

The sentence between lines 96 and 98 should be corrected, because it is meaningless, as well as the sentence on page 10 lines 355-357.

References are missing
in some places, e.g., page 3 line 97, page 4 line 175.

It is necessary to correct numerous grammatical and spelling errors in the text.

It is necessary to correct numerous grammatical and spelling errors in the text (e.g., italicise the names of plants and microorganisms; write the number separately from the unit)

Author Response

(The authors gave the same response as above.)

Round 2

Reviewer 1 Report

Almost all comments have been addressed by the authors. However, comment # 9 has not been fully addressed as follow: 

“The nanoparticles were measured to be between 23 and 168 nm in size”, and figure 6A: Since the shape of the NPs was not spherical, the measurement of size using zetasizer was incorrect."

Thus, the data related to the measurement of the size using zetasizer must be removed from the manuscript. 

Minor editing of English language is required

Author Response

Thank you for your thoughtful recommendation. In accordance with the recommendations provided, the aforementioned item (zetasizer) was eliminated.